

# Explain the Black Box for the Sake of Science:
# The Scientific Method in the Era of Generative Artificial Intelligence

Gianmarco Mengaldo[1,2]

[1]Department of Mechanical Engineering, National University of Singapore, 9 Engineering Drive 1, Singapore, 117575
[2]Department of Mathematics (by courtesy), National University of Singapore, 10 Lower Kent Ridge Road, Singapore, 119076

**Correspondence:** Gianmarco Mengaldo (mpegim@nus.edu.sg)

**Abstract.** The scientific method is the cornerstone of human progress across all branches of the natural and applied sciences, from understanding the human body to explaining how the universe works. The scientific method is based on identifying systematic rules or principles that describe the phenomenon of interest in a reproducible way that can be validated through experimental evidence. In the era of generative artificial intelligence, there are discussions on how AI systems may discover new knowledge. We argue that human complex reasoning for scientific discovery remains of vital importance, at least before the advent of artificial general intelligence. Yet, AI can be leveraged for scientific discovery via explainable AI. More specifically, knowing the 'principles' the AI systems used to make decisions can be a point of contact with domain experts and scientists, that can lead to divergent or convergent views on a given scientific problem. Divergent views may spark further scientific investigations leading to interpretability-guided explanations (IGEs), and possibly to new scientific knowledge. We define this field as Explainable AI for Science, where domain experts – potentially assisted by generative AI – formulate scientific hypotheses and explanations based on the interpretability of a predictive AI system. To support the argument, we go through a simple practical thought experiment in the Earth Sciences related to extreme weather. This discipline is particularly sensitive to the limitations of the argument we propose, and it allows us to draw some important conclusions and potential future directions, namely the need for causality and reproducibility, among others.

## 1 Introduction

Human progress is inherently related to our ability to observe the world around us, and make sense of it. This frequently means identifying a set of rules or principles that describe systematically the phenomenon we are interested in. The word 'systematic' is crucial here: it means that the rules identified generalize to observations of the phenomenon that were unseen when deriving those rules. Indeed, this is the cornerstone of the modern scientific method (Galilei, 1638), that has allowed us to discover new *knowledge* in various fields, from the natural sciences, including physics and biology, to the applied sciences, including medicine and engineering.

The scientific method is inherently connected to humans, as we have been the key players in discovering new *principles* (or theories) that explain the real world. However, with the advent of and recent advances in artificial intelligence (AI), machines are also learning 'principles' (intended in a colloquial meaning) on scientific problems that underlie natural phenomena – see





e.g., (Lam et al., 2023; Karagiorgi et al., 2022; Jumper et al., 2021; Schütt et al., 2019; Esteva et al., 2019; Novakovsky et al., 2023; Merchant et al., 2023; Yang et al., 2023). Today, for a machine to learn 'principles' means to identify patterns in data and learn relationships that may generalize to the problem the machine is set to solve. The identification of these principles (i.e., patterns and relationships) is usually achieved via defining a learning task for an AI model, training the model on that task, and verifying that the trained model generalizes to unseen observations. Once this step is achieved, we are commonly interested in

the results that the AI model gives us on the provided learning task, rather than on the 'principles' it used to reach those results. This is frequently satisfactory, and may be indeed our final goal. Yet, driven by curiosity, regulatory requirements, or else, we may want to know what 'principles' the machine used to obtain a certain result. To this end, the questions 'what' and 'why' are central to our discussion: (i) 'what' did the machine use to reach a certain decision, and (ii) 'why' did the machine use the 'what'? The 'what' can be the input-output relationships identified by the machine, or just the data deemed important by the

machine to reach a certain conclusion (when those explicit relationships are not available). In both cases, data is central. The 'why' is left to human complex reasoning to figure out, trying to interpret or explain the 'what'. We argue that, for answering the 'why' question, and uncover the principles learned by the machine, *data* is the key, and it represents the point of contact between machines and humans, at least for the natural and applied sciences. We name the *data* the machine deemed important to reach a certain result, the *machine view*, as depicted in Fig. 1. The scrutiny of the *machine view* by human scientists and

their complex reasoning can be the key to discovering new scientific knowledge. We shall make an important observation here: 'how' the machine used the information/data it is given is also an crucially important. However, that information is frequently not available to us, and deriving it may be quite challenging. Therefore, for this manuscript, we focus on the 'what' and 'why' questions, noting that the 'how' is equally important, when available.

To support our argument, we will dive into the scientific method that constitute the foundation of scientific discovery, and

show how this can be revisited in the era of AI through explainability. We will further present a thought experiment (yet practically reproducible) that shows how this could be achieved, while highlighting some potential limitations.

We should also add that we share the concerns regarding the risks of adopting AI for scientific research (e.g., (Krenn et al., 2022; Wang et al., 2023)) recently brought forward by Messeri and Crockett (Messeri and Crockett, 2024), where they state that: "adopting AI in scientific research can bind to our cognitive limitations and impede scientific understanding despite

promising to improve it". We believe that explainable AI (XAI) is a way of bridging the gap between human knowledge and machine's usage of the data, and can guide human experts to enrich human knowledge rather than being detrimental to it.

## 2   The scientific method in the era of Generative AI

AI is commonly seen as a black-box tool, whereby humans struggle to understand why the machine took a certain decision or provided a certain forecast. This prevents us to understand possible 'principles' AI may have learnt, that could be useful

to humans to enrich their knowledge or assess whether to trust the AI results. A way of addressing this issue, at least partly, is via XAI, where the latter is composed of two elements: (i) interpretability, and (ii) explanation of interpretability results in human-understandable terms; the two together provide a human-feasible pathway to 'explaining' AI, and yield interpretability-





guided explanations (IEGs). The first element, interpretability, aims to answer the question '*what*' (and possibly '*how*', when available (Rudin, 2019)) data the machine deemed important to reach a certain result. The second element, explanation of inter-

pretability results is left to domain experts and scientists that can generate hypotheses, if they have a divergent understanding of the data space compared to the machine. Answering the '*what*' and '*why*' questions can unveil those sought-for 'principles' the predictive AI model might have learnt.

The rationale for this argument is as follows. The scientific method, that underlies both the natural and applied sciences, is based on inductive reasoning, i.e., the formulation of hypothetical rules, and empirical evidence, i.e., observations and

experiments, that validate the hypothetical rules. The hypothetical rules may be formulated before or after empirical evidence, and they can in fact be amended and improved after new empirical evidence comes to light. This approach to the scientific method is closely related to the prevailing philosophical view of science provided by Popper in his work "*The Logic of Scientific Discovery*" (Popper, 2005), although alternative views exist – see e.g., Feyereband (Feyerabend, 2020), among others. To

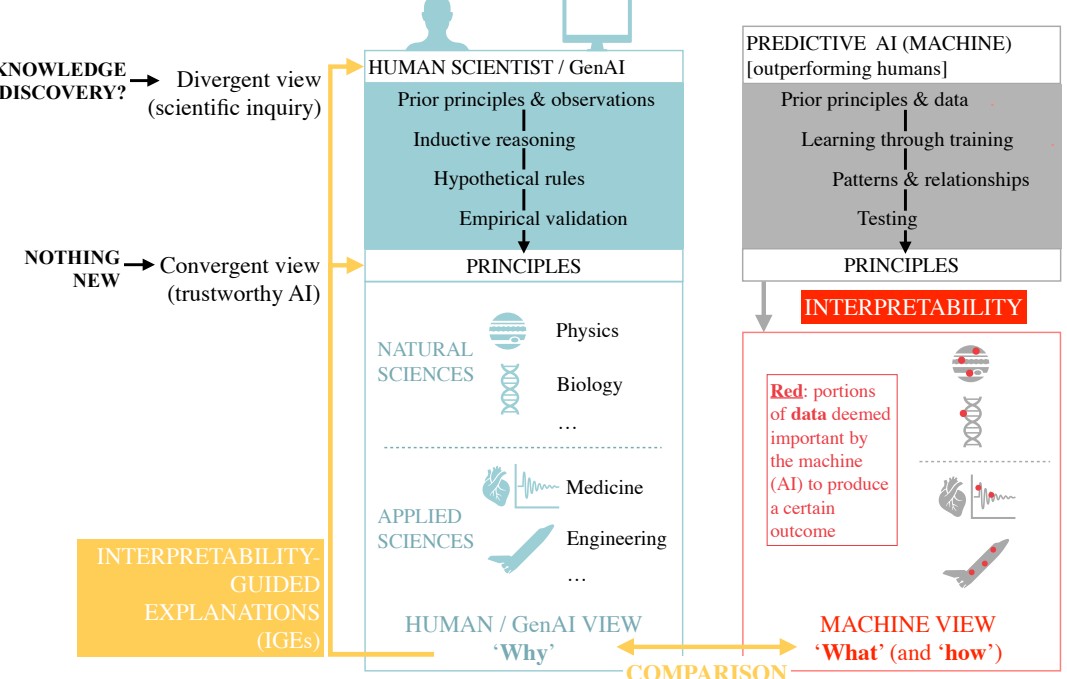

**Figure 1. The scientific method in the era of AI**. Comparative view of the approach used by humans (left, green) and AI (right, grey) to identify scientific principles that can generalize to unseen observations/data. Comparing the human view (existing knowledge) with the machine view (AI-used data) can drive new investigations from divergent perspectives or validate results for critical applications like medicine. The key to this process is interpretability (red box, bottom right), that can give us the machine view. Through this, we can anchor human explanations to the machine view, leading to interpretability-guided explanations (IEGs), that may lead to knowledge discovery.

understand this approach in practice, consider Newton's law of universal gravitation, which defines the force between two





masses as $F = Gm_1m_2/r^2$. Newton derived this law inductively from empirical observations (Newton, 1687), where the gravitational constant $G$ was only determined later through Cavendish's experiment.

Similarly, Einstein's theories of special (Einstein, 1905) and general relativity (Einstein et al., 1916), which extended Newtonian mechanics, were developed using both theoretical insights and empirical evidence. These examples highlight how scientific discovery often combines hypothesis formation, prior knowledge, and empirical validation.

This pattern appears across disciplines. In biology, Mendel deduced inheritance laws through pea plant experiments (Mendel and Mendel, 1970). In medicine, Fleming discovered penicillin by observing bacterial inhibition due to fungal contamination (Fleming, 1929). In engineering, Volta's battery was inspired by Galvani's findings on electrical stimulation in frogs.

These examples illustrate how scientific knowledge emerges from real-world observations and prior knowledge. Hypothetical rules are formulated, then validated or refuted through further evidence (Popper, 2005) – a process that underpins the
scientific method and drives discovery in the form of principles or theories (Fig. 1, left green box).

Observations, for what AI is concerned, are data and, in some cases, labels or known outputs, that the machine can use to learn a certain task, and become better at it by experience (i.e., training) (Mitchell et al., 1990). By training on some data (and labels), the machine learns some rules that are hopefully applicable to unseen data; in other words, the machine may have learnt some 'principles' in the form of patterns in data and relationships that generalize to unseen observations.

The 'principles' AI learns may or may not align with empirically validated scientific knowledge (Fig. 1, right red box). Often, we are satisfied with AI results without investigating why they were obtained – i.e., what 'principles' the machine used.

Two notable AI applications in science – weather forecasting (Lam et al., 2023) and protein folding (Jumper et al., 2021) – showcase this. Both fields have vast data and theoretical understanding. Despite this, AI has in some cases outperformed traditional methods based on human knowledge, suggesting it has identified underlying patterns that improve performance.
Hence, we might want to understand what those 'principles' are, and possibly close existing human-knowledge gaps. However, with the traditional workflow shown in Fig. 1 (right red block only) we are unable to achieve this step.

We argue that the key to bridge the gap between human and machine understanding is through interpretability methods applied to the AI system, that in turn can provide what we name here the *machine view* on a certain scientific problem (yellow box on the bottom right of Fig. 1). The *machine view* corresponds to '*what*' data the machine deemed important to obtain a
certain result, and it gives us a glimpse of those principles that the machine has learnt[1]. We can then compare the *machine view* with the existing body of knowledge available (what we name *human view*), and try to respond the '*why*' question, that is: why the machine deemed those data important. This comparison may lead to convergent views or divergent views on the problem being solved. Divergent views can spark further scientific investigation. More specifically, scientists and domain experts may look at the *machine view*, and take a divergent (yet machine viable) understanding on what is commonly accepted in a certain
field, thus providing possible pathways to fill knowledge gaps, and discover new knowledge.

---

[1]A glimpse and not the full picture, because to really retrieve those principles, we should respond to the '*how*' the machine used those data deemed important. To achieve this, we would need to have full access to the inner workings of the AI system, along with all the relationships among variables and parameters that constitute the model; route that is practically not viable today.



The workflow presented in Fig. 1 can open the path towards scientific discovery via XAI, also referred to as XAI for Science. The latter naming convention is in analogy with AI for science, and scientific machine learning, that usually embeds domain-specific knowledge into AI models to produce plausible domain-constrained results (Raissi et al., 2019). XAI for Science (also referred to as scientific XAI) may use both constrained and unconstrained AI models, where the data the machine used to take

a certain decision is available to the users.

However, in order for scientific XAI to have a chance of being successful, it needs to satisfy certain key pillars or requirements, that are necessary but not sufficient to achieve the task.

## 3 Explainable AI for Science: *what*, (*how*), and *why*

The starting point of XAI for Science is related to what we named the *machine view*, that answers the 'what' question: 'what' is

the data the machine deemed important to obtain a certain result? However, before diving into explainability and the machine view, we shall clarify a crucial assumption.

**Assumption.** *In this work, we assume that the predictive AI model has learnt something useful (i.e., some rules or 'principles' in the form of patterns and relationships) about the real-world phenomenon of interest. We further assume that those principles are encoded within the machine view. This translates into having a* causal AI model *that can accurately accomplish a scientific*

*task (e.g., predict the future weather evolution or a certain protein structure) similarly or better than state-of-the-art human-knowledge-driven models.*

The assumption just made implies that there is a chance of learning new rules in the form of patterns and relationships, using an AI model. In the assumption, we refer to a *causal model*. We note that the concept of causality, in machine learning,

and more generally in science, remains widely debated (Pearl and Mackenzie, 2018), and may take different connotations and viewpoints (Einstein et al., 1916; Bohr, 1937; Hume, 2000; Popper, 2005). Therefore, causality should be carefully assessed, by e.g., AI experts, as well as by scientists and domain experts, as part of their answer to the 'why' question.

Having framed our problem with the above assumption, we can now return to the *machine view*, and on how this can be retrieved. The relevant area of AI research for this task is called outcome interpretability, which explains how AI models use

input features to generate outputs (Yeung, 2020). The two main approaches within this context are: (i) post-hoc interpretability and (ii) ante-hoc interpretability.

Post-hoc interpretability (e.g., DeepLIFT (Shrikumar et al., 2017), SHAP (Lundberg and Lee, 2017), GradCAM (Selvaraju et al., 2017)) generate saliency maps (i.e., machine views) to highlight important input features (Samek et al., 2021). They are model-agnostic and do not alter AI models but may produce unreliable and non-unique explanations (Adebayo et al., 2018;

Slack et al., 2021; Rudin, 2019), requiring careful validation (Turbé et al., 2023).

Ante-hoc interpretability (e.g., decision trees (Letham et al., 2015), generalized additive models (Agarwal et al., 2021), prototype-based methods (Nauta et al., 2023)) may provide a more faithful machine view but may underperform in complex





tasks (Lam et al., 2023; Jumper et al., 2021). This trade-off remains a topic of debate (Rudin, 2019). For further reading on these methods, see (Rudin et al., 2022).

The issues just outlined pose challenges for our overall objective. Relying on inaccurate explanations can lead to scientific hallucinations (Messeri and Crockett, 2024), while non-reproducible or unclear insights may provide little value to scientists.

    To prevent such pitfalls, we define foundational pillars for scientific XAI, applicable to both post-hoc interpretability and self-explainable models. While various authors have proposed XAI requirements for different applications (Phillips et al., 2021; Rudin et al., 2022; Dwivedi et al., 2023; Cambria et al., 2023), we focus on three key pillars that the machine view should have

for XAI for Science to succeed, accuracy, reproducibility, and understandability, or in brief ARU:

1. **Accuracy.** The machine view should be accurate. In both post-hoc and ante-hoc interpretability, practitioners tend to assume that the machine view is accurate; assumption that is frequently wrong.

2. **Reproducibility.** The machine view should be reproducible, at least in a statistical sense, which means that if a certain
outcome explanation is given for a certain learning task, this can be systematically reproduced.

3. **Understandability.** The machine view must be understandable to human scientists and domain experts. This means that it should present viable features that could connect to existing knowledge. This requirement aligns with the common XAI desideratum of transparency, but here we emphasize its relevance to scientific understanding.

The first requirement is not new, and several frameworks have been deployed to evaluate interpretability results – see e.g., (Turbé et al., 2023) among others. The other two requirements are instead more tailored towards XAI for science, and specify how to interface the machine view to domain experts and scientists. We note that the important assumption on the model being causal made at the beginning of this section must hold (Beckers, 2022; Carloni et al., 2023). By having these pillars in place, scientists may grasp divergent views, increase their understanding of a given problem, and possibly fill knowledge gaps.

**A practical thought experiment in the Earth Sciences**. Let us now take a practical example to outline the workflow from the Earth Sciences, depicted in Figure 2. This consists of predicting if an extreme weather event (heatwave in the Indochina Peninsula) is occurring or not in the future (e.g., tomorrow or next week) given some spatio-temporal weather data available up to today (the thought experiment shown here is similar to the learning task presented in the preliminary work by Wei et al. (2025)). For the learning task, we use a Transformer neural network that we deem casual for the sake of showcasing the possible

workflow, and we setup a predictive classification task: classify if an extreme event is happening e.g., tomorrow/next week or not. The first step we perform is to obtain classification scores, and evaluate whether these scores are competitive, and the predictive AI outperforms state-of-the-art predictive skills in this context. If it does, we may then apply post-hoc interpretability, and assess that the relevance maps produced satisfy the ARU (accuracy, reproducibility, and understandability) requirements we just set. We then pass these interpretability results to human domain experts (meteorologists, and Earth system scientists),

and/or generative AI 'scientists' (large language models), and derive interpretability-guided explanations (also referred to as



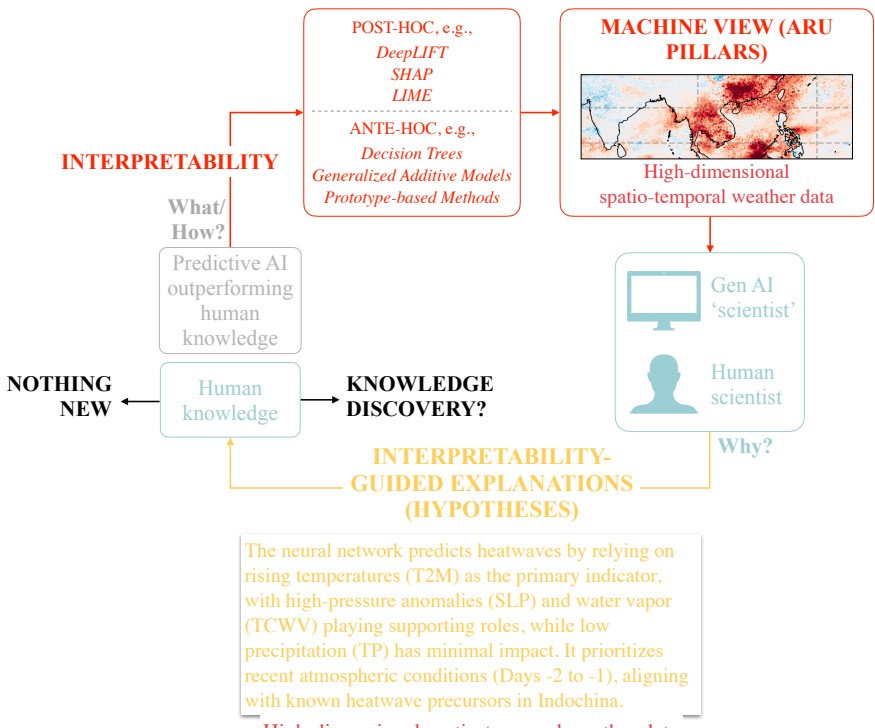

**Figure 2. XAI for Science workflow.** A predictive AI outperforms human expertise (grey box). Interpretability is then applied to generate the machine view (red boxes) that meet accuracy, reproducibility, and understandability (ARU) criteria. These interpretability results are then analyzed by human domain experts or generative AI models (green box) to derive interpretability-guided explanations (IGEs), which can either align with existing knowledge or lead to new scientific discoveries (in yellow). We depict one practical example in the context of the Earth Sciences related to extreme weather.

IGEs). These interpretability-guided explanations or IGEs can then be compared to existing human knowledge on the subject of extreme-weather precursors and they may: (i) generate nothing new, or (ii) yield knowledge discovery, as depicted in both Figures 1 and 2. The IGEs in the case of the yellow box in Figure 2 are generated via generative AI (i.e., ChatGPT); how to translate these explanations into actionable findings remains an open challenge, also showcasing the limitations of relying on

IGEs produced by generative AI. A promising avenue to address this challenge in the Earth sciences is through the discovery of equations related to poorly represented processes, as advocated by Huntingford et al. (2025), that can be aided by IGEs.

From this simple practical thought experiment, we shall note some clear limitations and potential issues of the proposed methodology, that require further AI developments. For instance: is the Transformer model really causal? What happens if we were to use a larger domain size for our classification task, thereby using data corresponding to a larger geographical area?

What if we were to use a set of different variables – would our results and interpretability maps change? These are critical questions that should be carefully considered, especially when dealing with scientific fields that are data rich, and for which we only posses partial knowledge, such as the Earth Sciences.



## 4  A cautionary tale

XAI for Science faces some obvious limitations, related to both the causality assumption, as well as to the ARU requirements
we set. In addition, translating the explanations provided by domain experts, and potentially by large language models, into
actionable findings is still a nascent field that demands significant developments over the next few years. We will go through
them, expanding on the example made at the end of the last section.

Let us start from the assumption of causality. The patterns and relationships identified by the machine may be just spurious
correlations, without any causality, and may fool scientists towards chasing wrong answers to the 'why' question. Understand-
ing whether causality is appropriate to the problem being investigated, and assessing whether the machine view is causal is a
key aspect that scientists should take into account in answering 'why' the machine deemed important certain data. Indeed, sev-
eral works are now exploring the intersection between XAI and causality (Carloni et al., 2023), two fields that in the computer
science community have drifted apart over the years (Beckers, 2022), but that should be kept together, especially in the natural
and applied sciences. To this end, some promising works are emerging – see e.g., (Schölkopf, 2022; Janzing et al., 2020; Xu
et al., 2020).

In terms of the accuracy requirement of the relevance maps, to obtain what we called the machine view there exist several
methods, grouped into two different categories, namely post-hoc and ante-hoc interpretability – see e.g., (Graziani et al., 2023)
for a comprehensive review and taxonomy of interpretable AI. Post-hoc interpretability suffers from inaccuracy, aspect that
may render futile the effort by domain experts to explain why certain data was used, if the data pinpointed is wrong. To this
end, accuracy evaluation methods are as important, see e.g., (Turbé et al., 2023; Wei et al., 2024). Intrinsic interpretability
may address the accuracy issues that plague post-hoc interpretability, yet they may be model-specific, human-constrained, and
unable to provide machine views for complex deployed models that may not be easily accessible. Yet, several promising works
are being carried out in the context of intrinsic interpretability – see e.g., (Alvarez Melis and Jaakkola, 2018; Zhang et al.,
2022; Turbé et al., 2024, 2025). Another recent promising area is (intrinsic) compositional interpretability (Tull et al., 2024),
that may help tie together different intrinsic interpretability viewpoints under a unified framework.

In terms of reproducibility, we should be able to retrieve the same machine view repeatedly, for a given task and set of
samples. In this context, some issues derive from the possible non-uniqueness of the machine view that interpretability methods
may provide (especially post-hoc interpretability methods). This can hinder our understanding of why those machine views
were used. To this end, answering the question '*how*' was the machine view (i.e., the data deemed important) used by the
machine can provide a more useful insight, as well pointed out by Rudin (Rudin, 2019).

In terms of understandability, the machine view should allow scientists and domain expert to tap into their body of knowl-
edge, to allow for comparing machine and human views on a certain phenomenon of interest. This is partly connected to what
discussed by Freiesleben et al. (Freiesleben et al., 2022). More specifically, the model should be able to address a question
understandable to scientists, with a machine view that can also be understood by scientists within their framework of domain-
specific knowledge. The choice of data and data configuration is therefore critical, and frequently arbitrary, leaving a degree of
subjectivity and arbitrariness to the problem.





Finally, how do we make potentially useful interpretability-guided explanations (IGEs) provided by domain experts action-able – in other words, how do we use them in practice for knowledge discovery? A key field that has consistently driven human innovations is our ability to synthesize mathematical models of the real world (Galilei, 1638). However, from IGEs to mathe-matical models there is still a significant gap, with symbolic regression (Angelis et al., 2023) and equation-based Earth system process discovery (Huntingford et al., 2025) possibly constituting feasible pathways.

Without a mathematical model, we may still grasp useful and actionable insights. For instance, Strogatz, in its inspiring New York Times editorial covering the winning of AlphaZero against Stockfish (Strogatz, 2018) states: "*What is frustrating about machine learning, however, is that the algorithms can't articulate what they're thinking. We don't know why they work, so we don't know if they can be trusted. AlphaZero gives every appearance of having discovered some important principles about chess, but it can't share that understanding with us.*". He additionally cites Garry Kasparov (the former world chess champion) that stated: "*we would say that its [AlphaZero] style reflects the truth. This superior understanding allowed it to outclass the world's top traditional program despite calculating far fewer positions per second.*" (Kasparov, 2018). A few years later from that editorial, researchers showed that Go players are improving their game looking at how AlphaGo, DeepMind's Go machine, is playing and providing them with a divergent view (Shin et al., 2023).

Science is a rather different application than Go (or Chess), and it is important to understand and assess carefully the limitations of XAI for knowledge discovery.

XAI is a possible path forward to keep complex human reasoning in the loop, possibly complemented by large language models and generative AI, while bridging current AI power in discovering patterns and relationships in data. XAI may also alleviate some of the risks that we may face when using AI for scientific discovery, that we share with Messeri and Crock-ett (Messeri and Crockett, 2024).



*Code and data availability.* A minimum code example and dataset for the results presented in Figure 2 will be made publicly available, although not central to the discussion.

*Author contributions.* Gianmarco Mengaldo conceptualized the work, and wrote the manuscript.

235 *Competing interests.* No competing interests.

*Acknowledgements.* We want to thank my two PhD students, Jiawen Wei and Chenyu Dong for many of the results presented in this work. We thank all members of my group at NUS, MathEXLab, for their dedication to research that made this work possible. We also thank all the colleagues that provided constructive criticism, from several different research fields and perspectives, that helped improve this manuscript significantly. We acknowledge funding from MOE Tier 2 grant 22-5191-A0001-0 'Prediction-to-Mitigation with Digital Twins of the Earth's 240 Weather' and MOE Tier 1 grant 22-4900-A0001-0 'Discipline-Informed Neural Networks for Interpretable Time-Series Discovery'.



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
