# Peer review of "Explain the Black Box for the Sake of Science: The Scientific Method in the Era of Generative Artificial Intelligence"

_EGUsphere, 2025_

## Referee Comment (RC1)

**Review of "Explain the Black Box for the Sake of Science: The Scientific Method in the Era of Generative Artificial Intelligence" by Gianmarco Mengaldo**

The paper discusses how the scientific method changes in the era of AI and explores opportunities offered by Explainable AI (XAI) for discovering new knowledge. It defines criteria for "XAI for science". It also points out some shortcomings of current XAI methods and mentions a few promising future developments.

I find the general idea of the paper, the re-interpretation of the scientific method in the era of AI, interesting and intriguing. The paper is well written, clear, it is fun to read. However, I do think that there are a couple of significant shortcomings of the paper – (1) the paper is too optimistic of what AI tools can deliver, (2) important aspects are not discussed in depth or are left out. Thus the discussion feels limited in depth. I expect more in-depth discussions from a paper having the impressive title "Explain the Black Box for the Sake of Science: The Scientific Method in the Era of Generative Artificial Intelligence" and coming up with the new concept of "XAI for science". This summarises basically my main criticism, I provide more details below.

**General comments**

1. I find the general message of the paper too optimistic considering the current abilities of the XAI tools and their pitfalls. This current pitfalls, however, do no mean that XAI cannot provide viable solutions in future scientific research. But then this is more a future goal instead of an assessment of the current state. The paper should place greater emphasis on the future development of XAI by offering recommendations and outlining possible trajectories to guide its role in science. What must be achieved to ensure a reliable 'XAI for science'?

2. The paper promises a lot, but fails to convince in case of certain crucial aspects. It says that it discusses the "what" and "why" questions ("how" seems to be out of reach because it is not discussed in the paper). However, the "what" question that can be answered currently (and that is answered in the example discussed) is only a "glimpse" (terminology used in the paper) of the actual "what" question, i.e. what principles has the machine learned and used to obtain certain results. The "why" question as it is now in the paper is misleading and should be rephrased. By comparing the machine view with the human / GenAI view (as illustrated in Fig 1) one cannot answer "why the machine deemed those data important", as phrased in the paper, but instead only "why **we** (or GenAI) **think** that the machine deemed those data important". This difference is crucial. The first version suggest that we can indeed find out why the machine deemed those data important, even when we use only post-hoc interpretability methods, whereas the latter version expresses clearly that we can only come up with **new hypothesis** based on XAI methods. These hypothesis have to be then verified and proven, as it is done in the classical scientific process. As it is formulated currently in the paper, the why question is actually equivalent with the "how" question, which is not discussed in the paper.

3.  The paper states that by comparing interpretability-guided explanation with existing human knowledge, it may: (i) generate nothing new or (ii) yield new knowledge. There is however a third option: **generate false "knowledge"**. Although mentioned in the last section, this third option is not stated here (L 166-167). It should be stated here as well for the sake of completeness.

4.  Divergence between the machine view and the human / gen AI view can lead to new scientific knowledge. It can lead however also to false "knowledge". How to decide whether we should trust the new "knowledge" or not? What strategies, techniques could we use? This is not discussed in the paper, although it is a crucial aspect.

5.  Also related to the point above, the author writes that the AI results "should present viable features that could connect to existing knowledge". I think this should be explained better. The new knowledge might not connect well to existing knowledge. How to differentiate between right and false new knowledge especially when the knowledge found by the machine does not connect well with the existing knowledge?

6.  The author states that "XAI may also alleviate some of the risks that we may face when using AI for scientific discovery, that we share with Messeri and Crockett (Messeri and Crockett, 2024)." I would be interested in how exactly XAI could alleviate the illusions of "explanatory depth", "exploratory breadth" and "objectivity"? I can even think of ways XAI strengthening these illusions: XAI offers an explanation, which might suggest there is no reason for more in-depth analysis (illusions of "explanatory depth"), XAI is an AI method, thus it rules out hypotheses not testable with AI (illusion of "exploratory breadth"), it suggests objectivity, but it is data specific (illusion of "objectivity"). While the author states "we share the concerns" of Messeri and Crockett these are not further examined, and XAI is instead suggested as a remedy, which may seem like an overly simplistic response to a complex challenge. It would provide more depth to the paper if it would discuss this issue in more detail.

7.  The paper defines foundational **pillars for scientific XAI**: accuracy, reproducibility, understandability. While I agree with the importance of these concepts, the author does not discuss how these criteria could be assessed and quantified. Related to the described example, the author writes that after applying post-hoc interpretability, one can "assess that the relevance maps produced satisfy the ARU requirements", but then the discussion of the example stops with just providing these maps. I think the most important step is missing from this example:

    a.  What decision do we take based on the provided relevance map and how? Is this finding "nothing new", "new knowledge" or a spurious result?

    b.  Are the ARU requirements satisfied and how to check that?

Going deeper into these questions, also using the example provided in the paper, would make the paper more useful for the Earth Science community, from a practical perspective.

**Specific comments**

1. Related to the translation to actionable knowledge: from current AI methods we mainly obtain "hints" for hypotheses regarding principles, and then humans or generative AI can generate possible interpretations. Some comments on possible errors in this last step could be useful.

2. **Generalisability** of data-driven findings is different from generalisability in sense of the classical scientific method, i.e. mathematical and physical principles. AI models are trained using a subset of the data, and if the model learns well, its skills are generalisable to another part of the data on which the model was not trained on. However, this data has still very similar characteristics with the training data. If the model is applied then to a different dataset the "principles" might not be valid anymore, at least fine-tuning is needed. Thus, this generalisability is at a lower level compared with what is possible in the classical scientific method, when we find general mathematical and physical principles which we can apply to all possible datasets and unsees. I would appreciate it if the paper would at least shortly discuss this difference.

3. L. 27 "Identification of these principles" suggest that the principles are identified by humans, which is not the case. It should be pointed out that the identification is meant from the perspective of the machines.

---

## Referee Comment (RC2)

I commend Gianmarco Mengaldo's efforts in addressing what is a rapidly developing issue within (Earth) science – the role of artificial intelligence. "Explain the Black Box for the Sake of Science: The Scientific Method in the Era of Generative Artificial Intelligence" is a reasoned and serious attempt at addressing some of the opportunities and risks arising from the rapid ascent of a new generation of AI tools.

Given the subject matter, the paper could be deeply philosophical. e.g What differentiates 'the scientific method' from other epistemic frameworks? What is the scientific method? What is science? Depending on discipline, school of thought, or dogma, very different answer could be supplied. Mengaldo's flavour of Popperianism would, arguably, be widely shared within the Earth science community. Latourian's may well disagree. For the purposes of this paper I would say let them. But it needs to be acknowledged that there is not a single, agreed understanding of science and this may matter given the subject matter. Indeed, it is my opinion that it does matter for this manuscript as I argue below.

More generally, and philosophical issues aside, great care needs to be taken in defining terms. How, what, why are used in the manuscript as centrally important concepts. How and why could become muddled. To help avoid that, I would suggest a (clearer) treatment of teleology is provided. Or at least, let us hear what are Mengaldo's assumptions are on the matters of intentionality when it comes to both the scientific method and the operation of particular AI algorithms.

It would also be useful to clearly state the particular types of AI that are exclusively discussed. Perhaps in an alternative universe in which AI research was not entirely dominated by large language models and other connectionist algorithms and (often very large) data sets, the very basis of Mengaldo thesis would be moot. We would not need some sort of intermediary scientific process in order to 'understand' AI algorithms, because symbolic approaches would be – in some respects by design – much more transparent. But we do not live in that universe.

I found it odd that there was not more of a discussion about the nature of the AI algorithms and some of the mathematical basis for connectionist approaches. I do not think it is safe to assume that Earth system scientists have such knowledge. Indeed, unless I have misunderstood, the problem that Mengaldo seeks to address is that connectionist approaches

produce black box algorithms because the approximation of functions via connection interactions/weights means that not only can we not use them to render a model that 'is as simple as possible, but no simpler', but the way the algorithm operates in a very high dimensional space means it can be utterly incomprehensible to humans. The latest generations of large transformer models have billions of parameters. The structures in data these algorithms are finding or creating doesn't 'mean' anything to us. Hence the need for XAI.

Mengaldo's ambition is to use insights from *how* algorithms are using *what* data to produce specific outputs. This isn't just a sensitivity analysis. I understand the motivation here to address that deep epistemic issue outlined immediately above. If one adopts an instrumentalist stance to science then perhaps' Mengaldo's task becomes a tiny bit less Herculean. All algorithms are models, all models are wrong, some are more useful. We can define utility in a number of ways. This could short-circuit any convoluted discussions around causation. I was not very convinced by how that concept was treated in this manuscript. How is (non)linear regression associated with causation? Does it matter? Mengaldo does indeed refer to some relevant literature, but I think this raises more questions than it answers.

Mengaldo discusses accuracy, reproducibility, and *understandability*. It is understandability that I think Mengaldo wishes to address with XAI. Connectionist models are sometimes already doing a better job than process-based models with respect to some Earth science with regards accuracy. Reproducibility is certainly an issue when it comes to non-analogue conditions, when (empirical) data begins to significantly move away from the training data (e.g. non-analogue models of climate). But there is then the issue of how do they do what they do? That may really be incomprehensible, but could we still glean important *information* from these algorithms? Important information in this respect would be how this information could be used to inform *process-based* models of the phenomena of interest. The example Mengaldo uses discusses temperature and/or precipitation.

This understanding matters because comprehension is an important element of trust. Yes, an AI may reliably produce highly accurate output, but if we cannot understand how it has used certain data and how resilient that algorithm is to different data, then we may feel limited in our abilities to trust it. In my more pessimistic moments I wonder if at some point in the not

too distant future, such concerns will be considered as quaint. The raw computational power and size of connectionist models will means highly accurate and reliable outputs will be produced. A constant supply of miracles before breakfast. As to the question of the how, why bother? In that respect, induction would have won. These models outputs are true because they are always right. Black swans be damned.

Perhaps underneath these discussions there is an issue of comprehension or in some sense tractability. Mengaldo begins the discussion with Newton's famous law of gravitation. One way of telling the history of science is that we started with the easy problems – the low hanging and then falling fruit when it comes to gravity – and have more recently been struggling with complexity. It may be the case that much of the natural phenomena that we are interested in simply cannot be described using such elegant formalism. Algorithms like backpropagation can be shown to – given certain conditions – approximate polynomial functions. Understanding in terms of formalism can be preserved. But what these models are effectively doing is producing polynomials that have so many terms, are so complex that we cannot relate them to any processes – we cannot understand them. XAI is motivated to bridge that gap, but it may prove too profound a chasm to scale. In effect, we would have replaced one complex difficult to understand system (a natural phenomenon) with an engineered system that ultimately proves as *effectively* as complex and difficult to understand. What of scientific progress then? Does it stop?